# Study on Quantum Radar Detection Probability Based on Flying-Wing Stealth Aircraft

**DOI:** 10.3390/s22165944

**Published:** 2022-08-09

**Authors:** Shaoze Lu, Zhijun Meng, Jun Huang, Mingxu Yi, Zeyang Wang

**Affiliations:** School of Aeronautic Science and Technology, Beihang University, Beijing 100191, China

**Keywords:** detection probability, stealth aircraft, quantum radar, QRCS

## Abstract

The development of quantum radar technology presents a challenge to stealth targets, so it is necessary to study the quantum detection probability. In this study, an analytical expression of the quantum radar cross section (QRCS) for complex targets is presented. Based on this QRCS expression, a calculation method for the detection probability for quantum radar is creatively proposed. Moreover, a self-designed flying-wing stealth aircraft is adopted to obtain the detection probability distributions of the conventional radar and the quantum radar in different directions. As revealed by the result of this study, the detection probabilities of the quantum radar and the conventional radar are significantly different, and the detection probability of the quantum radar has obvious advantages in most regions with a certain distance.

## 1. Introduction

Quantum radar technology has been recognized as an advanced detection technology combining radar technology and quantum technology. This technology breaks through the performance limits of typical detection using the quantum characteristics of electromagnetic waves [1,2,3,4]. The appearance of quantum radar has presented a severe challenge to stealth targets, so the study of the scattering characteristics of quantum radar and the quantum detection probability of targets has become an important field.

In the study on the quantum radar detection of a target, Marco Lanzagorta [5,6,7,8] initially proposed the concept of quantum radar cross section (QRCS) in accordance with the theory of quantum electrodynamics. Lin et al. [9] proposed a QRCS calculation method applicable to plate-shaped targets and numerically verified the calculation results. The above numerical method exhibits high computational efficiency and is capable of calculating the QRCS of an ideal conductor plate with arbitrary shapes. Liu et al. [10,11] developed quantum radar equations and investigated the QRCS of corner reflectors under single-photon incidence. Fang et al. [12,13] found and analyzed the difference between the QRCS and RCS of a cube in the main lobe. They suggested that the above novel macroscopic quantum phenomenon could be adopted to detect and identify stealth weapons and in the field of biomedicine. Chen et al. [14] explored the single-photon and two-photon QRCS of a single surface under the cylindrical model. Two-dimensional targets have been studied in depth, thus laying a solid theoretical and numerical basis for the above research, while the research on targets’ QRCS has primarily focused on the numerical calculation of typical targets.

In a study on the radar detection probability of a target, Wu et al. [15] obtained the dynamic RCS sequence based on the coordinate system transformation between the radar and the target and built a discovery probability model of the search radar. The effect of the characteristics of the incoming target and the search radar parameters on the radar discovery probability was analyzed using the models of an F-16 fighter and BGM-10 cruise missiles. Liu et al. [16] developed a feasible method to assess the stealth performance of aircraft in the key angle region by integrating the mobile smoothing algorithm and a detection probability model. The relative fluctuation error was reduced using the moving smoothing algorithm to process RCS data, and the radar detection probability was obtained to assess the detectable performance of aircraft.

Although great progress has been made in the quantum scattering properties and the conventional radar detection probability of some targets, researchers have also investigated the detection probability of conventional radar in depth. However, quantum radar research on stealth aircraft and other complex targets has been rare. Quantum radar detection probability of complex targets shows great development potential.

The contribution of this study is an optimized formulation derived to calculate the QRCS of a complex stealth target, in which the photon parameters, the occlusion factor, and the vector dot product are adopted. Compared with the previous approaches, the proposed method ensures the accuracy of calculation, and is applicable to most targets such as flying-wing aircraft. Meanwhile, by introducing the quantum signal-to-noise ratio and the quantum radar cross section, this study creatively proposes a calculation method of the quantum radar detection probability applicable to complex targets, and explores the effect arising from a single variable on such detection probability. In addition, we design a flying wing stealth aircraft, and obtain the detection probability distributions of conventional radar and quantum radar, respectively, in the horizontal direction and 4π direction. It is found that the quantum detection probability has obvious advantages in some specific regions.

## 2. Theory of Detection Probability on Quantum Radar

### 2.1. The Concept and Simplified Expression of the Quantum Radar Cross Section

Starting from the interaction mechanism between quantum radar and target, the definition of the scattering cross section of the quantum radar is given, based on [5]:(1)$$\sigma_{Q}≜\underset{R\to \infty }{\lim}4\pi R^{2}\frac{\langle {\hat{I}}_{s}\left(r_{s},r_{d},t\right)\rangle }{\langle {\hat{I}}_{i}\left(r_{s},t\right)\rangle }$$
where ${\hat{I}}_{i}(r_{s},t)$ and ${\hat{I}}_{s}(r_{s},r_{d},t)$ denote the intensity of the incident field and scattering field, respectively; $r_{s}$ and $r_{d}$ represent the position of the transmitter and receiver of the quantum radar (monostatic radar $r_{s}=r_{d}$), respectively; and R expresses the distance between the radar and target. For more specific information, please refer to [17].

The Cartesian coordinate system is defined for any point p=(x,y,z) on the target surface. $\theta_{i}$, $\theta_{s}$ are the angle of the incident wave and scattered wave, respectively. $\phi_{i}$, $\phi_{s}$ represent the angle between the projection of the incident wave and the scattered wave at plane $O_{XY}$ and the axis *X*.

For any convex target, the analytical expression of its quantum radar cross section (QRCS) $\sigma_{Q}$ is written as [8]
(2)$$\sigma_{Q}=4\pi A_{⊥}(\theta_{i},\phi_{i})\frac{{\left|\sum_{n=1}^{N}e^{ik\Delta R_{n}}\right|}^{2}}{\int_{0}^{2\pi }\int_{0}^{\pi }{\left|\sum_{n=1}^{N}e^{ik\Delta R_{n}}\right|}^{2}\sin\theta_{s}d\theta_{s}d\phi_{s}}$$
where $A_{⊥}(\theta_{i},\phi_{i})$ denotes the orthogonal projected area of the target in each incidence and $\Delta R_{n}$ is total interference distance. $\Delta R_{n}=\Delta R_{n}^{t}+\Delta R_{n}^{r}$ where $\Delta R_{n}^{t}$ denotes the distance from a transmitter to an atom on the surface of an object and $\Delta R_{n}^{r}$ represents the distance from the atom to the receiver. It yields:(3)$$e^{ik\Delta R_{n}}=e^{ik\left(\Delta R_{n}^{t}+\Delta R_{n}^{r}\right)}$$

The incident photon wave vector is assumed to point from a distance to the origin of the coordinates, and the reflected photon wave vector is assumed to point away from the origin of the coordinates. $k_{i}$ denotes the incident wave vector of the photon and $k_{s}$ represents the scattered wave vector of the photon.

Substituting $k_{i}$ and $k_{s}$, it yields:(4)$$e^{ik\Delta R_{n}}=e^{ik_{i}\cdot d_{T_{x}}+ik_{s}\cdot d_{R_{x}}+i(k_{i}-k_{s})\cdot p_{n}}$$
where $p_{n}$ denotes the position vector of the *n*th atom. Since $k_{i}\cdot d_{T_{x}}$ and $k_{s}\cdot d_{R_{x}}$ are not dependent on the position of the atoms, they can be taken out before the summation. With photon parameters ${K=k}_{i}-k_{s}$, (2) is written as:(5)$$\sigma_{Q}=4\pi A_{⊥}(\theta_{i},\phi_{i})\frac{{\left|\sum_{n=1}^{N}e^{iK\cdot p_{n}}\right|}^{2}}{\int_{0}^{2\pi }\int_{0}^{\pi }{\left|\sum_{n=1}^{N}e^{iK\cdot p_{n}}\right|}^{2}\sin\theta_{s}d\theta_{s}d\phi_{s}}$$

The summation in (5) is transformed into an integral form, and its Fourier form is given. If the incident wave of the photon irradiates a certain point on the target surface, while diffraction is not considered, the scattered wave of the photon on the above point only exists in the normal space of the tangent plane. The occlusion factor function $V\left(\theta_{s},\phi_{s},p_{n}\right)$ is defined to represent the integral region:(6)$$V\left(\theta_{s},\phi_{s},p_{n}\right)=\left\{\begin{array}{c} 1,scattered\;waves\;from\;p\;can\;travel\;in\;the\;direction\;of{\;k}_{s}\;\\ 0,scattered\;waves\;from\;p\;can’t\;travel\;in\;the\;direction\;of{\;k}_{s}\; \end{array}\right.$$

For a 3D target, its normal vector at the point $p_{n}$ is defined as $n_{p}$. It yields:(7)$$V\left(\theta_{s},\phi_{s},p_{n}\right)=\left\{\begin{array}{l} 1,n_{p}\cdot k_{i}{\;<\;0\;\mathrm{and}\;n}_{p}\cdot k_{s}\;>\;0\\ 0,n_{p}\cdot k_{i}{\;>\;0\;\mathrm{or}\;n}_{p}\cdot k_{s}\;<\;0 \end{array}\right.$$

Introducing the surface density of atoms m, i.e., the number of atoms per unit area, it yields:(8)$$\sum_{n=1}^{N}e^{iK\cdot p_{n}}=\iint e^{iK\cdot p}V\left(\theta_{s},\phi_{s},p\right)mdS$$

Setting $F\left(V\left(\theta_{s},\phi_{s},p_{n}\right)\right)=\iint e^{iK\cdot p_{n}}V\left(\theta_{s},\phi_{s},p_{n}\right)mdS$, it yields:(9)$$\sigma_{Q}=4\pi A_{⊥}(\theta_{i},\phi_{i})\frac{{\left|F\left(V\left(\theta_{s},\phi_{s},p_{n}\right)\right)\right|}^{2}}{\int_{0}^{2\pi }\int_{0}^{\pi }{\left|F\left(V\left(\theta_{s},\phi_{s},p_{n}\right)\right)\right|}^{2}\sin\theta_{s}d\theta_{s}d\phi_{s}}$$

Equation (9) is the Fourier transform of (5). 

When dealing with targets, it is cumbersome to use numerical integration, and the finite element method is employed for calculation. The conversion process from direct integral solution to finite element solution is presented below. 

The surface should fall into triangular facets. $n_{m}$ represents the normal vector of plane element. $V_{a_{m}},V_{b_{m}}$ denote the vectors on both sides of the triangle, thus leaving the area of the surface element:(10)$$A_{m}=\frac{1}{2}\left|V_{a_{m}}\right|\left|V_{b_{m}}\right|\sin\theta_{m}=\frac{1}{2}\left|n_{m}\right|$$

$p_{m}$ denotes the center of the element. Combining (5) and (7), (9), (10), and $p_{m}$ together, it yields
(11)$$\sigma_{Q}=4\pi A_{⊥}(\theta ,\phi )\cdot \frac{{\left|\sum_{m=1}^{M}A_{m}e^{iK\cdot p_{m}}\right|}^{2}}{\int_{0}^{2\pi }\int_{0}^{\pi }{\left|\sum_{m=1}^{M}A_{m}e^{iK\cdot p_{m}}\right|}^{2}\sin\theta_{s}d\theta_{s}d\phi_{s}}$$
where *M* denotes the number of surface elements of the target surface. Equation (11) does not consider occlusion. If occlusion is considered, $V_{m}$ should be substituted into (11). Subsequently, (11) is expressed as:(12)$$V\left(\theta_{s},\phi_{s},p_{n}\right)=\left\{\begin{array}{l} 1,n_{m}\cdot k_{i}{\;<\;0\;\mathrm{and}\;n}_{m}\cdot k_{s}\;>\;0\\ 0,n_{m}\cdot k_{i}{\;>\;0\;\mathrm{or}\;n}_{m}\cdot k_{s}\;<\;0 \end{array}\right.$$
(13)$$\sigma_{Q}=4\pi A_{⊥}(\theta ,\phi )\cdot \frac{{\left|\sum_{m=1}^{M}V_{m}(\theta_{i},\phi_{i},p_{m})\cdot A_{m}e^{iK\cdot p_{m}}\right|}^{2}}{\int_{0}^{2\pi }\int_{0}^{\pi }{\left|\sum_{m=1}^{M}V_{m}(\theta_{i},\phi_{i},p_{m})\cdot A_{m}e^{iK\cdot p_{m}}\right|}^{2}\sin\theta_{s}d\theta_{s}d\phi_{s}}$$
where
(14)$$A_{⊥}(\theta ,\phi )=-\frac{1}{2}\sum_{m=1}^{M}V_{m}(\theta_{i},\phi_{i},p_{m})\frac{n_{m}\cdot k_{i}}{\left|k_{i}\right|}$$

Equation (13) represents the final simplified expression of 3D target’s QRCS.

### 2.2. Expression of Detection Probability Combined with Quantum Radar

In this section, to obtain the detection probability formula of the quantum radar, the signal-to-noise ratio (SNR) formula needs to be used first, which refers to [18].

The above formula comprises the received power $P_{R}$ and the noise equivalent power NEP:(15)$$SNR_{Q}=\frac{P_{R}}{NEP}$$

First, the received power $P_{R}$ adopted by the conventional radar is discussed.

To obtain the received power of the quantum radar, the undirected power density of incident photons is expressed as:(16)$$\langle I_{i}\rangle =\frac{1}{2\eta }{\left(\frac{\varepsilon_{0}^{*}}{R}\right)}^{2}e^{-\Gamma (t-R/c)}$$
(17)$$\varepsilon_{0}^{*}=-\frac{\xi \omega^{2}}{4\pi \varepsilon_{0}c^{2}\Delta r_{id}}$$
where ω denotes the angular frequency; Γ represents the reversal life of the excited state of the atom; η is the quantum efficiency; $\varepsilon_{0}$ is the dielectric constant; $\Delta r_{id}$ expresses the distance between the ith atom and radar; ξ denotes the quadratic power of the vector dot product; and R is the distance between the target and radar. Γ=0, $\Delta r_{id}=R$, and it yields:(18)$$\langle I_{i}\rangle =\frac{1}{2\eta }{\left(\frac{\varepsilon_{0}^{*}}{R}\right)}^{2}$$

In accordance with the definition in Formula (1) of the quantum radar scattering cross section, the scattering intensity is solved as:(19)$$\langle I_{s}\rangle =\frac{\langle I_{i}\rangle \sigma_{Q}}{4\pi R^{2}}=\frac{\left({\left(\varepsilon_{0}^{*}\right)}^{2}/2\eta \right)\sigma_{Q}}{4\pi R^{2}}=\frac{\varepsilon_{0}^{*2}\sigma_{Q}}{8\pi R^{4}\eta }$$

Multiply both sides by the radar aperture area to obtain the received power:(20)$$P_{R}=\langle I_{s}\rangle A_{r}=\frac{\varepsilon_{0}^{*2}A_{r}\sigma_{Q}}{8\pi R^{4}\eta }$$

After relevant parameters are given, the QRCS Formula (13) derived in this study is substituted into (20) to determine the value of received power, thus laying a basis for the calculation of SNR and quantum detection probability.

For the noise of the quantum radar, since the bulk noise is the main aspect, NEP can be replaced by the bulk noise, which is expressed as:(21)$$\eta_{\sigma }=\sqrt{\frac{2{\left(h\nu \right)}^{2}\Delta \nu Ne^{-\chi R}}{\eta }}$$

Where χ denotes the loss factor; ν represents the photon frequency; Δν expresses the noise bandwidth; and N is the number of target atoms. Substituting (20) and (21) into (15), it yields:(22)$$SNR_{Q}=\frac{\langle I_{s}\rangle A_{r}}{\eta_{\sigma }}=\frac{{\left(4\pi \varepsilon_{0}^{*}\right)}^{2}A_{r}\sigma_{Q}}{2{\left(4\pi \right)}^{3}R^{4}\eta }\sqrt{\frac{\eta }{2{\left(h\nu \right)}^{2}\Delta \nu Ne^{-\chi r}}}$$

For the detection probability, the Gram-Charlier series and the detection probability formula of Swerling type targets used [19,20] are expressed as:(23)$$P_{D}=\frac{erfc(V/\sqrt{2})}{2}-\frac{e^{-V^{2}/2}}{\sqrt{2\pi }}[C_{3}(V^{2}-1)+C_{4}V(3-V^{2})-C_{6}V(V^{4}-10V^{2}+15)]$$
where constants C_3_, C_4_, and C_6_ represent the coefficients of the series, respectively as:(24)$$C_{3}=-\frac{SNR+1/3}{\sqrt{n_{p}}{\left(2SNR+1\right)}^{1.5}}$$
(25)$$C_{4}=\frac{SNR+1/4}{n_{p}{\left(2SNR+1\right)}^{2}}$$
(26)$$C_{6}=C_{3}^{2}/2$$

Variable V is:(27)$$V=\frac{V_{T}-n_{P}(1+SNR)}{\varpi }$$
(28)$$\varpi =\sqrt{n_{p}(2SNR+1)}$$
where $n_{p}$ is the cumulative number of pulses and $V_{T}$ is the detection threshold.

By substituting the derived SNR formula into (23), the detection probability formula of the quantum radar can be obtained:


(29)
$$\begin{array}{l} P_{DQ}=\frac{1}{2}erfc\left(\frac{V_{T}\eta_{\sigma }-n_{p}\left(\eta_{\sigma }+\langle I_{s}\rangle A_{r}\right)}{\sqrt{2n_{p}\eta_{\sigma }\left(2\langle I_{s}\rangle A_{r}+1\right)}}\right)-\frac{1}{\sqrt{2\pi }}\exp\left(\frac{\left(V_{T}\eta_{\sigma }-n_{p}\left(\eta_{\sigma }+\langle I_{s}\rangle A_{r}\right)\right)2}{2n_{p}\eta_{\sigma }\left(2\langle I_{s}\rangle A_{r}+1\right)}\right)\left[-\frac{\langle I_{s}\rangle A_{r}\sqrt{\eta_{\sigma }}+\eta_{\sigma }^{3/2}/3}{\sqrt{n_{p}}{\left(2\langle I_{s}\rangle A_{r}+\eta_{\sigma }\right)}^{1.5}}\left(\frac{{\left(V_{T}\eta_{\sigma }-n_{p}\left(\eta_{\sigma }+\langle I_{s}\rangle A_{r}\right)\right)}^{2}}{n_{p}\eta_{\sigma }\left(2\langle I_{s}\rangle A_{r}+1\right)}-1\right)\right.\\ +\frac{\frac{\langle I_{s}\rangle A_{r}}{\eta_{\sigma }}+1/4}{n_{p}{\left(2\frac{\langle I_{s}\rangle A_{r}}{\eta_{\sigma }}+1\right)}^{2}}\left(\frac{V_{T}\eta_{\sigma }-n_{p}\left(\eta_{\sigma }+\langle I_{s}\rangle A_{r}\right)}{\sqrt{n_{p}\eta_{\sigma }\left(2\langle I_{s}\rangle A_{r}+1\right)}}\right)\left(3-\frac{{\left(V_{T}\eta_{\sigma }-n_{p}\left(\eta_{\sigma }+\langle I_{s}\rangle A_{r}\right)\right)}^{2}}{n_{p}\eta_{\sigma }\left(2\langle I_{s}\rangle A_{r}+1\right)}\right)-\frac{{\left(\langle I_{s}\rangle A_{r}\sqrt{\eta_{\sigma }}+\eta_{\sigma }^{3/2}/3\right)}^{2}}{2n_{p}{\left(2\langle I_{s}\rangle A_{r}+\eta_{\sigma }\right)}^{3}}\left(\frac{V_{T}\eta_{\sigma }-n_{p}\left(\eta_{\sigma }+\langle I_{s}\rangle A_{r}\right)}{\sqrt{n_{p}\eta_{\sigma }\left(2\langle I_{s}\rangle A_{r}+1\right)}}\right)\cdot \\ \left.\left(\frac{{\left(V_{T}\eta_{\sigma }-n_{p}\left(\eta_{\sigma }+\langle I_{s}\rangle A_{r}\right)\right)}^{4}}{4n_{p}^{2}\eta_{\sigma }^{2}{\left(2\langle I_{s}\rangle A_{r}+1\right)}^{2}}-\frac{10{\left(V_{T}\eta_{\sigma }-n_{p}\left(\eta_{\sigma }+\langle I_{s}\rangle A_{r}\right)\right)}^{2}}{2n_{p}\eta_{\sigma }\left(2\langle I_{s}\rangle A_{r}+1\right)}+15\right)\right] \end{array}$$


In addition, to compare with the quantum detection probability, this study also calculates the conventional radar detection probability of the stealth aircraft at different azimuth angles [21].

The total power transmitted from the target signal to the radar is written as:(30)$$P_{R}=\frac{P_{t}G^{2}\lambda^{2}\sigma A_{r}}{{\left(4\pi R^{2}\right)}^{2}}$$
(31)$$G=\frac{4\pi A_{r}}{\lambda^{2}}$$
where $P_{t}$ is the transmitted power, λ is the wavelength, G is the gain, and σ is the RCS of the conventional radar.

Substituting (31) into (30) yields:(32)$$P_{R}=\frac{P_{t}A_{r}^{2}\sigma }{4\pi R^{4}\lambda^{2}}$$

The noise power of the lossless antenna is written as:(33)$$N=kT_{e}B$$
where k is the Boltzmann constant; B is working bandwidth; and $T_{e}$ is the effective noise temperature.

The signal-to-noise ratio of the conventional radar is:(34)$$SNR_{C}=\frac{P_{R}}{NFL}=\frac{P_{t}A_{r}^{2}\sigma }{4\pi kT_{e}BFLR^{4}\lambda^{2}}$$
where F is the noise coefficient and L is radar loss.

The detection probability of the conventional radar can be obtained by substituting (34) into (23).

## 3. Research Methods

In this study, the CATIA modeling software is first adopted to design and draw a high stealth aircraft. CATIA refers to a mainstream software integrating CAD, CAE, and CAM, and it has been extensively applied in aerospace, automobile, and other industries [22,23]. Figure 1 illustrates the modeling of the flying-wing stealth aircraft drawn in this study.

As depicted in Figure 2, the length, wingspan, and height of this type of flying-wing stealth aircraft are 1670 mm, 2060 mm, and 170 mm, respectively. The sweeping angles of leading edge and trailing edge of the wing reach 56 degrees and 36 degrees, respectively. Since the stealth aircraft does not have significant bulges, the calculation method of the quantum radar cross section and quantum detection probability deduced in this study can be applied to it. The radar feature of the flying-wing stealth aircraft is that the intensity of RCS at different azimuth angles and detection probability is significantly different. Additionally, the reason for the choice of this flying-wing stealth aircraft is to better explore the detection probability characteristics and differences between conventional radar and quantum radar.

In this study, the effect arising from a single variable on the detection probability is investigated, and the above detection probability formula and stealth aircraft converted into a grid (Figure 3) are imported into MATLAB based on the flying-wing stealth aircraft. Subsequently, the detection probability of different azimuth angles in the horizontal direction, as well as the detection probability and average value of the 4π direction, are obtained for the aircraft. Figure 2 illustrates the calculation method of the detection probability in the 4π direction. Figure 2 shows that detection probability of each circumferential direction of the aircraft is calculated respectively by controlling different vertical angles to obtain the changing characteristics of the detection probability of the conventional radar and the quantum radar.

Furthermore, the calculated azimuth is 0~360°, the vertical angle is −90~90°, and the step size is 1°.

## 4. Simulation Results 

### 4.1. Verification for a Typical 3D Target

In the present section, several numerical examples based on single-photon incidence are selected to indicate the efficiency and accuracy of the proposed algorithm.

The accuracy of the calculation equation derived in this study should be verified before the complex objective optimization. Accordingly, this optimized algorithm and the method of the reference in this section are adopted to obtain the QRCS of two typical 3D convex targets. The accuracy and calculation time of the above algorithms are compared.

The first example is a cylindrical target (Figure 4). The radius of the cylinder is 0.3 m, the height of the cylinder is 0.5 m, and the incident wavelength is 0.25 m.

Figure 5 presents the verification of the simplified QRCS expression of the cylinder. The figure indicates a high consistency between the proposed method and the method in [24], where “before improvement” represents the results achieved using the method [24], and the “after improvement” represents the results achieved using the method of this study. As depicted in the figure, the data acquired using the optimized algorithm applied in this study are very consistent with the calculation result of [24].

The above results suggest that the QRCS calculation method presented is confirmed to be feasible and accurate in solving the typical cylindrical target QRCS. 

Figure 5 illustrates the time achieved by different formulations for various grid scales.

### 4.2. Parameters’ Influence on Detection Probability of Quantum Radar

In the present section, we discuss the effect arising from different parameters on the quantum radar and the conventional radar, and preliminarily understand the effect of the above variables on detection probability by changing one or two related variables where the variables of frequency, RCS (QRCS), and distance exhibit high commonality in the two radars, and the above variables are also important factors influencing the detection probability of the quantum radar. Accordingly, this section focuses on the above parameters.

#### 4.2.1. Effect of Frequency on Detection Probability

The first part of the present section discusses the effect of frequency (Figure 6, Figure 7 and Figure 8). Figure 6 illustrates the correlation between the detection probability and frequency in the case of multiple RCS and QRCS. The dotted line in the figure represents the detection probability of the quantum radar, while the marked solid line represents the detection probability of the conventional radar. The values of RCS and QRCS are 0.01 m^2^, 0.1 m^2^, 1 m^2^, and 10 m^2^.

In the following coordinate graph, for the convenience of observation, a part of the detection probability is expressed by dB, and its conversion relation is expressed as:(35)$$P_{D}\left(dB\right)=10\times \log_{10}P_{D}$$

As depicted in Figure 6a, in the RCS and QRCS certain cases, the goals of the quantum radar and conventional radar detection probability are within a certain range. The detection probability rapidly increases with the increase in the frequency, then it shows a slow growth and tends to be 100%. The quantum radar and conventional radar detection probability can be indicated with frequency as an index. The shape of the respective curve is basically similar. However, when the detection probability tends to 0%, it is difficult to see the changing trend of the above curves. 

The difference between the detection probability of the two radars is that RCS and QRCS decrease, and the curve change of the quantum radar is faster than that of the conventional radar. For instance, when RCS and QRCS are equal to 0.01 m^2^, the two detection probability curves (red) are close to coincidence, while at 10 m^2^, the conventional detection probability curve (green) is significantly lower. Thus, when RCS and QRCS are large, the quantum radar can more effectively find targets.

In Figure 6b, due to the detection probability in the graph expressed in the dB, the target in the trend of the detection probability is observed to be close to 0. The above two types of detection probability curve in a low frequency range have an obvious steep slope. Subsequently, the slope then begins to slow down, shows a short linear increase, and finally reaches 0 dB. 

Moreover, when RCS and QRCS drop from 10 m^2^ to 0.01 m^2^, the curve shows a significant right shift. In other words, if a certain detection probability is ensured, the smaller the RCS and QRCS, the higher the frequency will be required.

Figure 7 shows the change of detection probability at multiple pulse accumulative numbers (*n_p_*). As depicted in Figure 7a, both the quantum radar and the conventional radar have the same curve variation trend. In both cases, the curve keeps moving to the left when the pulse accumulation number keeps increasing. In other words, when the detection probability remains unchanged, the required frequency becomes lower. 

Figure 7b shows that when the frequency is close to 0.1 GHz, the quantum radar and the conventional radar converge to the dB of the same detection probability when the pulse accumulation number is constant. 

For instance, when *n_p_* = 5, both radars maintain the position of −276 dB under the condition of being close to 0.1 GHz; when *n_p_* = 500, both are at −94 dB. 

With the increase in the pulse accumulative number, the dB values of near-convergent detection probability at 0.1 GHz for both of them increase significantly.

Figure 8 depicts the change of detection probability under multiple false alarm probabilities (*P_fa_*). It can also be observed that, when the frequency approaches 0.1 GHz, the dB values of the detection probabilities of the two radars also rise significantly with the increase in the false alarm probability *P_fa_*, especially when the frequency is significantly low; the convergence position of the detection probability is significantly improved.

#### 4.2.2. Effect of Detection Probability between the Distance of Target and Radar

The second part is the effect of radar relative target distance on detection probability (Figure 9, Figure 10 and Figure 11). Figure 9 describes the correlation between detection probability and non-detection probability with frequency in the case of multiple RCS and QRCS. The dashed line still represents the detection probability of the quantum radar, while the solid line represents the detection probability of the conventional radar.

As depicted in Figure 9, the detection probability of the quantum radar and the conventional radar on targets gradually decreases with the increase in detection range, and the above curves converge at the position of −480 dB. However, the difference between them is that, in Figure 9b, when the distance tends to increase, the detection probability of the quantum radar drops faster at the corresponding position, dropping from close to 100% to close to 0% in a relatively short distance interval. 

Moreover, when RCS and QRCS decrease, the amplitude by which the quantum detection probability curve moves to the left is smaller, so the influence on the detection probability of the quantum radar and target detection range is smaller than that of the conventional radar. 

For instance, when RCS and QRCS are 10 m^2^, the quantum detection probability is lower, while the probability of detection is significantly higher than that of the conventional radar at 0.01 m^2^.

The detection probability curves in Figure 10 and Figure 11, respectively, compare false alarm probability and frequency. As depicted in the figure, the detection range of the two radars decreases with the increase in false alarm probability and becomes longer with the increase in frequency.

#### 4.2.3. The Effect of QRCS and RCS on Detection Probability

The third part (Figure 12, Figure 13 and Figure 14) studies the variation characteristics of the detection probability curve when the independent variables are QRCS and RCS.

By the change of the curve in Figure 12, we found that the RCS and QRCS aperture with radar has a great impact on the conventional detection probability and the quantum detection probability. The two kinds of detection probability are increased with the increase in the aperture area, and the variation in the QRCS’s influence on quantum detection probability is smaller compared with the effect of the RCS on the conventional detection probability.

Figure 13 and Figure 14, respectively, show the correlation between detection probability and RCS and QRCS at different distances and frequencies. It can be found in Figure 13 that, when RCS and QRCS are below 0.001 m^2^, the detection probability shows obvious convergence.

At the same time, with the decrease in the distance, the quantum detection probability gradually exceeds the conventional detection probability. When the distance is 1000 km, the probability of the conventional radar is slightly higher, while when the distance is 480 km, the probability of conventional detection is significantly lower than that of the quantum radar. Accordingly, the detection probability of the quantum radar for short distances is more advantageous than the conventional detection probability. The phenomenon in Figure 14 also shows a positive correlation between the quantum detection probability and the frequency.

As indicated by the research results in this section, the above factors significantly affect the detection probability. 

It can be seen that the detection probability of the conventional radar and the quantum radar is positively correlated with RCS, QRCS, frequency, and aperture area, and negatively correlated with distance, and that the lower limit of the two detection probabilities increases with the increase in pulse accumulation number and false alarm probability. The above findings can also provide a certain reference for the following stealth aircraft detection probability simulation.

### 4.3. Quantum Detection Probability of a Stealth Aircraft in the Horizontal Plane and with Conventional Radar

In the present section, we conduct a simulation study on the detection probability of traditional and quantum radar at different azimuth angles of a flying-wing stealth aircraft without pitch angle (see Figure 15). 

Through this work, this study explores the distribution characteristics of the quantum and conventional detection probability of the stealth aircraft at different angles, and also discovers and analyzes the differences and advantages of the quantum detection probability compared with the conventional detection probability. The simulation process is shown in the figure below.

In this section, three types of parameter variables are set and will be studied around the above variables. For variables that are not involved, the default values of the arguments include: a frequency of 5.3 GHz, a range of 1000 km, a radar aperture area of 13.9 m^2^, a quantum efficiency of 0.62, a pulse accumulation of 2, a false alarm probability of 10^−9^, a loss factor of 2.21 × 10^−5^, a target atom number of 8.43 × 10^27^, and a noise bandwidth of 8.6 × 10^8^.

The default values of the conventional radar parameters are: transmitting power 27.5 kW, noise temperature 298 K, working bandwidth 60 Hz, noise coefficient 5.7 dB, and radar loss 7.6 dB.

#### 4.3.1. Distance

In this part, six variables ranging from 600 km to 1800 km are set for the effect arising from target relative radar distance on the detection probability of the stealth aircraft.

At the same time, we calculated RCS and QRCS of this type of stealth aircraft as a comparison. QRCS used for quantum radar was calculated using the formula derived in this study, and RCS used for the conventional radar was calculated using the physical optics method.

Figure 16 depicts the distribution characteristics of the RCS and QRCS of the flying-wing stealth aircraft. For the conventional radar, the RCS of the stealth aircraft peaks at four azimuth angles, 36° and 124°. The above angles coincide with the sweep angle of the wing, so the leading and trailing edges of the stealth aircraft become the main radar scattering sources.

However, for quantum radar, the figure above does not show obvious peak values of QRCS, but shows a relatively uniform zigzag shape. Thus, in the detection of targets by quantum radar, the leading edge and trailing edge of the wing of the stealth aircraft are no longer the main scattering sources.

The six graphs in Figure 17 respectively represent the comparison of the detection probability of the quantum radar and the conventional radar at different distances. The detection probability of both is expressed in dB, and a value of detection probability lower than −500 dB can be ignored in the figure below.

As depicted in the above figure, the detection probability of both radars tends to decrease with the increase in the distance. From 600 km in Figure 17a to 1800 km in Figure 17f, the detection probability (P_D_) of this type of stealth aircraft with the conventional radar reveals that the detection probability is significantly high at four angles close to the front and rear edges of the wing, and the detection probability is low in the forward and tail directions. The detection probability of the front and rear edges of the wing decreases slightly only after the distance reaches 1800 km, and the angle range is slightly narrowed. In the 30° range of the forward and stern directions, the probability does not exceed the peak value of −100 dB, even at the highest detection probability at 600 km.

In addition, the detection probability close to 90° and 270° laterally is higher; the intensity is second only to the four specific directions, and the range of variation in the above distance is also the largest. The lateral probability is close to 0 dB at the smallest distance and drops below −200 dB at the furthest distance of 1800 km.

For the detection probability (*P_DQ_*) of the stealth aircraft quantum radar, the detection probability curve at 600 km reaches the top at the majority of the azimuth angles, slightly decreases until 1000 km, and basically drops below −300 dB at 1800 km. It is therefore revealed that, with the increase in the distance, the quantum detection probability decreases rapidly at the respective azimuth angle. It can also be seen that its distribution in different azimuth angles is more uniform than the conventional detection probability, and there are few obvious peaks. Accordingly, its detection capability at all azimuth angles, in the absence of pitch angle, can be considered approximately equal.

In comparison, the quantum detection probability under the above conditions is much higher than that of the conventional radar, except that the quantum detection probability is lower than that of the conventional radar in a certain angle range, and the dB values in the more important head and tail directions are much higher than that of the conventional radar. In the lateral case, the detection probability is lower than that of the conventional radar only when the distance is 1800 km, and the other five distances have obvious advantages.

By comparing the above probability distribution maps with the distribution characteristics of RCS and QRCS, it is not difficult to see that the detection probability of the two radars is closely related to RCS and QRCS. For instance, in the conventional radar, the detection probability of the above azimuth angles is significantly high due to the peak value of RCS at the leading edge and trailing edge angles. In the lateral position, due to the scattering of the lateral surface of the stealth aircraft and the edge of the wing, the probability of lateral RCS and detection is also high, while the probability of other angles is significantly low. For the quantum radar, since the leading edge and trailing edge of the wing have little influence on QRCS and the scattering source mainly lies in the surface of the fuselage, its QRCS does not have significant peak values at different azimuth angles, thus resulting in similar values of quantum detection probability at different angles.

Meanwhile, Table 1 shows the average value of detection probability at all azimuth angles. It can be seen that, at 600 km, the mean detection probability of the quantum radar is close to 0 dB, far greater than that of the conventional radar. Starting from 1200 km, this mean value is significantly lower than that of the conventional radar, and with the increase in distance, the mean detection probability of the quantum radar decreases faster. Thus, quantum radar is more outstanding in finding targets at a relatively short distance. Moreover, if the distance is long, the conventional radar is relatively better.

#### 4.3.2. Radar Aperture Area

In this study, six variables ranging from 1.4 m^2^ to 21.5 m^2^ were set to observe the effect arising from aperture area on detection probability. Since RCS and QRCS are not affected when the aperture area and distance are changed, there is no need to present the QRCS diagram in this part.

From the figure above, it can be seen that the dB values of the conventional and quantum detection probability of the stealth aircraft both rise with the increase in the radar aperture area. For the target detection of the conventional radar, we find that it still has a high detection probability at several specific angles, which is significantly affected by the aperture area. For instance, its peak value is only higher than −200 dB when the aperture area is 1.4 m^2^, and reaches the top of polar coordinates (0 dB) when it reaches 7.3 m^2^. However, for quantum radar, it is less affected by the aperture area. Although there is an obvious decrease in Figure 18a,b, the quantum detection probability in (a) is much larger than the conventional detection probability in all azimuth angles, so its decrease is smaller than that of the conventional radar. Although, from (a) to (f) in Figure 18, the two detection probabilities have an upward trend on the whole, the increase is mainly from 1.4 m^2^ in (a) to 7.3 m^2^ in (c). However, if the area continues to increase after that, the change of detection probability will be relatively gentle. Accordingly, in this case, it does not make much sense to continue to increase the radar aperture area.

Table 2 lists the mean detection probabilities of the quantum radar and the conventional radar with different aperture areas. As depicted in this table, the quantum and the conventional radar in the aperture area of 1.4 m^2^ and the average detection probability in the aperture area of 2.6 m^2^ are much lower compared with the other conditions, and the two average values increase along with the aperture area, first increasing suddenly, then rising. The characteristics of the quantum radar is that the main difference between the degree of the change of the average detection probability is smaller than the conventional radar. Thus, the use of adequate aperture area is still vital to the detection capability of the quantum radar and the conventional radar.

#### 4.3.3. Frequency

In this part, seven variables ranging from 1.1 GHz to 10 GHz are set to observe the effect arising from frequency on the detection probability of the stealth aircraft.

Figure 19 calculates the RCS, QRCS, and detection probability of each azimuth angle, respectively. Observation from Figure 19a–g shows that RCS and QRCS change with the increase in frequency, which leads to irregular changes in the quantum detection probability and the conventional detection probability at different azimuth angles and continued rising overall. For instance, under the condition of 8.1 GHz, the detection probability of the conventional radar in the tail is still relatively large, but it decreases after 10 GHz. For the quantum detection probability, there are small peaks at the azimuth angles of 90° and 140° at 2.9 GHz, but they do not exist after the frequency increases. Accordingly, it is further verified that the detection probability at different azimuth angles will be significantly affected by RCS and QRCS.

In addition, the curve of the quantum detection probability varies with frequency to a greater extent than that of the conventional detection probability. At the lowest frequency of 1.1 GHz, the detection probability of the quantum radar basically drops to the minimum value; it exceeds that of the conventional radar at 5.5 GHz; and it has basically reached 0 dB at 10 GHz. Thus, in this case, the frequency change has a greater impact on the target detection performance of the quantum radar.

Table 3 calculates the mean values of the two detection probabilities. The feature shown in this table is that, with the increase in frequency, the detection probability of the quantum radar and the conventional radar keeps increasing, while the quantum detection probability increases faster; and conversely, it decreases faster. At 1.1 GHz, its detection probability is already much lower than that of conventional radar. Accordingly, under such conditions, quantum radar requires a higher frequency to ensure its search ability, while the conventional radar requires a lower frequency.

As revealed by the experiment in the present section, the distribution characteristics of the detection probability of the quantum radar and the conventional radar at different azimuth angles are closely related to their respective QRCS (RCS). This is further verification of the quantum radar and the conventional radar detection probability, in the case of different variables.

Furthermore, when the frequency and radar aperture area are sufficiently high, the distance between target and radar is short and the vertical angle is low, the detection probability of the quantum radar is more advantageous than that of the conventional radar. In addition, the quantum detection probability also has a more obvious advantage in most azimuth angles such as the head and tail.

### 4.4. Detection Probability in the 4π Direction of the Quantum Radar and Conventional Radar for Stealth Aircraft

In this study, a quantum radar detection probability simulation (4π direction) for all azimuth angles is performed to explore the distribution characteristics of the detection probability at different altitude angles and azimuth angles. In this section, the quantum and conventional radar detection probability of the flying-wing stealth aircraft in the 4π direction are investigated in two aspects.

#### 4.4.1. Distance

Figure 20, Figure 21 and Figure 22 depict the 3D graph of the probability distribution of the target with a distance of 1000~1800 km in the 4π direction of two different radars, a cloud graph, and the pie graph of the percentage of dB values of different detection probabilities in the cloud area, respectively.

As depicted in Figure 20, Figure 21 and Figure 22, the probability distribution of omnidirectional quantum detection is significantly different from that of conventional detection. Under the conditions of the three, the conventional detection probability in the location of the poles, which is close to 90° and 90° vertical angle on the height, all reach the top (red area). The value of angle on the edge of the wing area is also significantly high, and the area of greater than −1 dB in the cloud covers a wide range; the highest reaches 41.70%, and the lowest reaches 29%. However, in the vertical angle of less than 30°, the two directions of the aircraft drop to nearly −400 dB (blue area), and the detection probability dB values of the two areas are significantly different. Furthermore, with the increase in the vertical angle, the detection probability will rise sharply, and the required altitude angle from −400 dB to 0 dB should only be nearly 30°.

The main difference between the three graphs is that the blue area expands towards the poles from 10.96% to 32.10% with the increase in the distance. The red area of the four azimuth angles close to the sweep angle is only slightly reduced, and the difference attributed to the change of distance is insignificant. As revealed by the above phenomena, the surface of the stealth aircraft and the vertical angle of radar irradiation on the target significantly affect the conventional detection probability, especially when it is close to ±90°. When the vertical irradiation impacts on the fuselage, it is easy to reflect electromagnetic waves, thus significantly increasing the possibility of detection.

As revealed by the distribution of the detection probability of the quantum radar, when the distance is 1000 km, the percentage of the area larger than −1 dB in the cloud image reaches 32.10%. Although the area of such area is not as large as that of the conventional radar, it accounts for over 54% in −1~−100 dB, so it exceeds that of the conventional radar when the distance reaches over −100 dB. In addition, the dark red area of the conventional detection probability is concentrated in the position with a high vertical angle, while the quantum detection probability is greater in the area above −100 dB, even in the area with a low vertical angle of −20° to 20° compared to that with a high vertical angle. Thus, in the above region, except for the four wing edge azimuth angles, most positions are more advantageous compared to the conventional detection probability.

After the distance tends to increase to 1800 km, the omnidirectional detection probability of the quantum radar decreases significantly. Moreover, the proportion larger than −1 dB remains only 1%, and the proportion of −1~−100 dB also decreases to 8%, largely distributed at both ends of the cloud map. In the vertical angle of −20°~20°, it basically decreases to below −300 dB. In the vertical angle of 20°~80°, the blue area increases significantly, whereas it remains close to 0 dB. The detection probability here remains higher than at the altitude angle of −20°~20°.

The difference between this and conventional detection probability lies in its probability distribution, which does not vary significantly with the change of azimuth angle. Although the quantum detection probability is maximum at a vertical angle of 90°, it does not vary with the vertical angle of the conventional radar like a cliff. It is not difficult to find that the quantum detection probability is less affected by vertical angle than that of conventional detection.

Furthermore, the quantum radar’s cross-scattering area for a target with an inclination angle is stronger than that of the conventional radar, i.e., the change of cross-scattering area for the radar is smaller than that of the conventional radar when the included angle between the surface and an incident photon increases or decreases. Accordingly, the quantum detection probability will not increase or decrease in a stepped manner due to a significant change of the vertical angle, thus resulting in the higher detection probability of the quantum radar compared to that of the conventional radar in the range of short distances, small vertical angles, and most azimuth angles.

Table 4 lists the mean values of the probabilities of conventional and quantum detection in the 4π direction at different distances. 

It is therefore revealed that, although the mean quantum detection probability at 1800 km is significantly smaller than that of the conventional radar, the quantum detection probability still shows an advantage when the distance decreases to 1000 km, thus verifying that quantum radar has a higher detection probability at a shorter distance.

#### 4.4.2. Aperture

Figure 23, Figure 24 and Figure 25 depict the 4π direction probability distribution of a stealth aircraft target with an aperture of 0.6~7.3 m^2^ for two different radars.

The above graphs show that the probability of most regions in the above graphs tended to increase with the increase in the aperture area. When the area is 0.6 m^2^, the conventional detection probability is basically dark red above the vertical angle of 70°, and the area of this area reaches 14.81% of the total area. It displays a cliff-like decrease below 70°, even at the four peak azimuth angles, so it is deep blue (less than −400 dB) in most areas with lower vertical angles, taking up 65.48%. When the area reaches 7.3 m^2^, the area of dark red increases significantly, the area of less than −1 dB accounts for 33.65%, and the area of −1~−100 dB takes up 17.61%. Although the area of dark blue decreases to 21%, the probability that the vertical angle is close to 0° remains significantly low.

For the quantum detection probability, when the area is 0.6 m^2^, the probability of detection is lower than −1 dB, only accounting for 1.94%. Although the size of the quantum radar is lower than that of the conventional radar on the whole, in most regions below 70°, most of the probability ranges from −1 to −200 dB, accounting for 31.83% and 21.63% of the total, respectively. Thus, when the altitude angle is low, the detection ability of the quantum radar is still dominant.

Under the area of 7.3 m^2^, the detection probability rises as a whole in most areas, and the detection probability of areas larger than −1 dB accounts for 18.34%. Although this is not as high as that of the conventional radar, much of this is distributed at a relatively low altitude angle, and the proportion of areas between −1 dB and −100 dB accounts for 58.16%. Compared with the conventional detection probability in the same case, the probability of the vertical angle above 50° is lower. However, when it is below 50°, the performance advantage of the quantum radar is more significant since most cases are higher than −100 dB.

Table 5 lists the mean of the 4π direction detection probability for different aperture areas. As depicted in the table, at 0.6 and 2.7 m^2^, the probability of the conventional radar in the area of the vertical angle higher than 60° and less than −60° remains significantly high, so its mean value exceeds that of the quantum radar. However, after 7.3 m^2^, the situation is also reversed. Moreover, in the case of small area, the effect on the mean of the omnidirectional detection probability of the two radars is significant, thus revealing the significance of aperture area for the above two radars. Only when the aperture area is larger can the detection ability be excellent.

In the present section, the distribution characteristics of the detection probability of two radars at different vertical angles and azimuth angles are explored and analyzed by drawing 4π direction cloud maps and 3D coordinate maps. It is found that the quantum radar is better than the conventional radar in finding stealth targets in most areas with a vertical angle of ±30°.

## 5. Conclusions

In this study, a new formula for the quantum radar cross section is first derived, which ensures the calculation accuracy compared with the existing work. A method is proposed to obtain the detection probability of the quantum radar, and the effect arising from different variables on detection probability is explored, as well as two radar detection probability distribution characteristics in the horizontal direction and the 4π direction. Based on the above experiments, the following conclusions are drawn:The detection probability of the conventional radar and the quantum radar is positively correlated with RCS, QRCS, frequency, and radar aperture area, while being negatively correlated with distance.Since the characteristics of the QRCS and RCS of the stealth aircraft at various azimuth angles are significantly different, the detection probabilities of the quantum radar and the conventional radar at different azimuth angles are significantly different. Additionally, the difference of the quantum detection probability at different altitude angles is smaller than that of the conventional detection probability.When the distance between target and radar is short and the vertical angle is low, the detection probability of the quantum radar is more advantageous than that of the conventional radar. In contrast, when the distance between target and radar is longer, and the vertical angle is higher, the conventional radar shows more advantages in target detection.For the flying-wing stealth aircraft, the quantum detection probability at 0–360° azimuth angle is significantly different from that of the conventional radar. The conventional detection probability peaks at the angle perpendicular to the edge, and the difference between different azimuth angles is large, while there is a slight difference between the quantum detection probability at different azimuth angles.

## Figures and Tables

**Figure 1 sensors-22-05944-f001:**
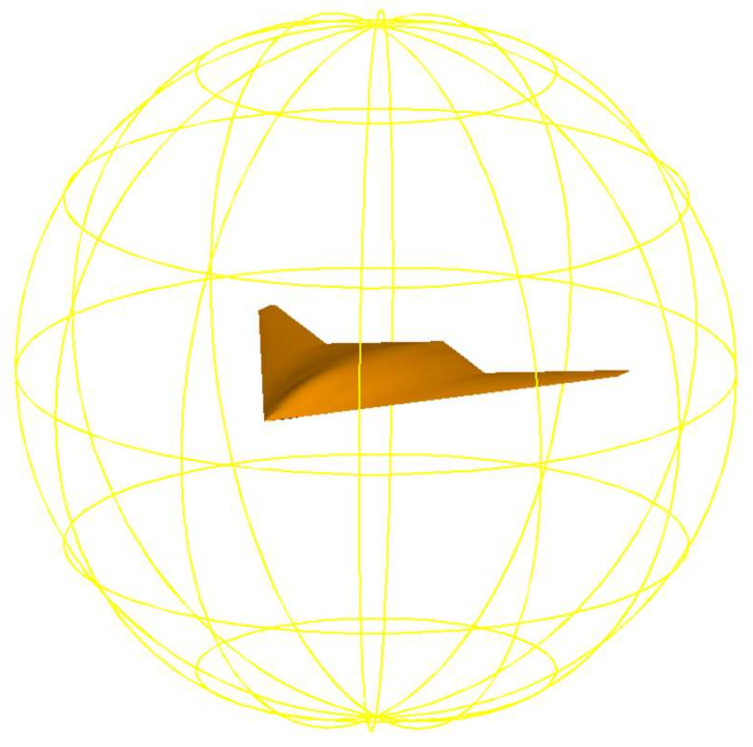
Calculation of the probability of detection in the 4π direction.

**Figure 2 sensors-22-05944-f002:**
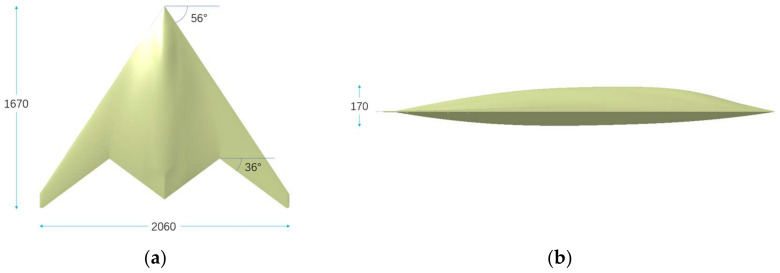
Geometry of a stealth aircraft. (**a**) Vertical view, (**b**) Side view.

**Figure 3 sensors-22-05944-f003:**
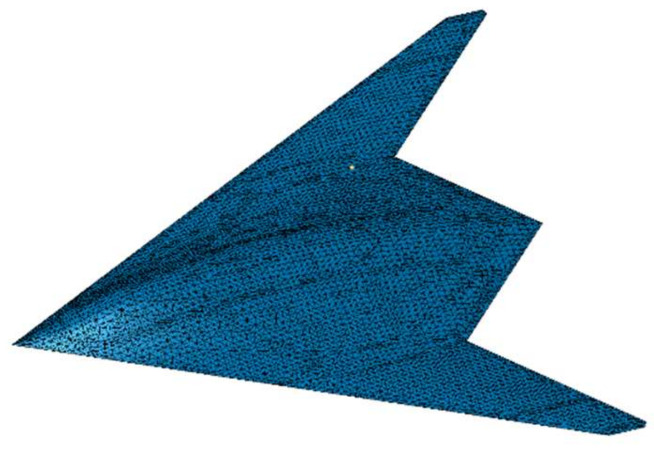
The grid of a stealth aircraft.

**Figure 4 sensors-22-05944-f004:**
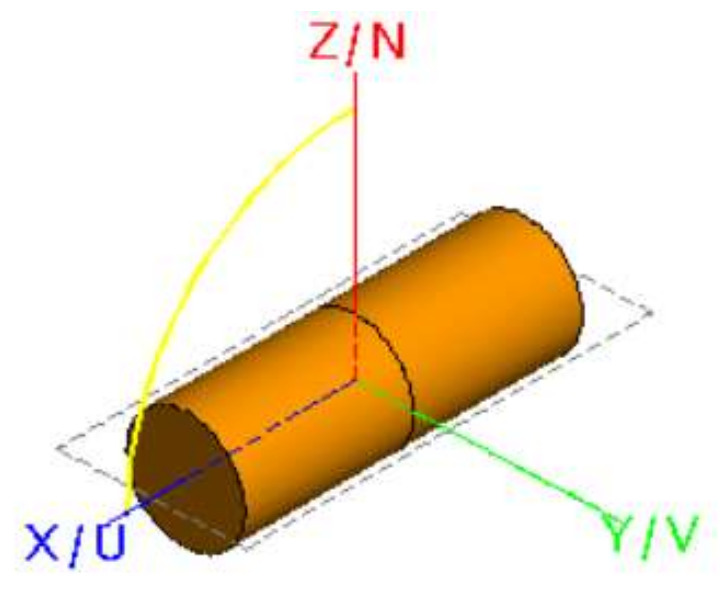
Geometry of a cylinder.

**Figure 5 sensors-22-05944-f005:**
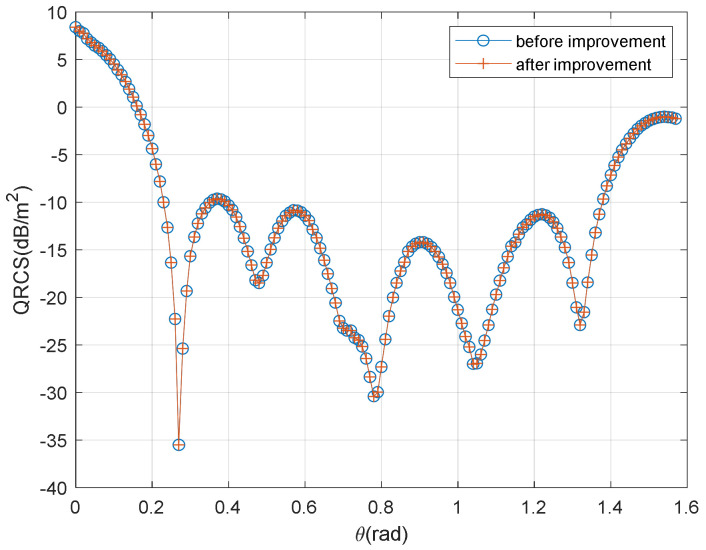
Verification of the simplified QRCS expression for cylinder.

**Figure 6 sensors-22-05944-f006:**
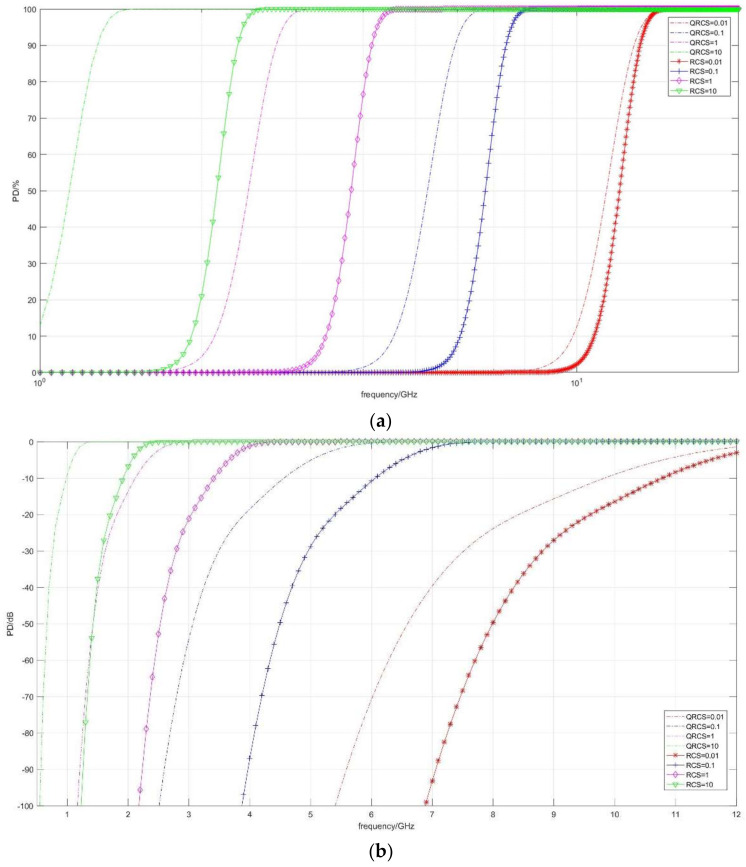
Correlation diagram of detection probability changing with frequency (taking RCS and QRCS). (**a**) *P_D_*/%, (**b**) *P_D_*/dB.

**Figure 7 sensors-22-05944-f007:**
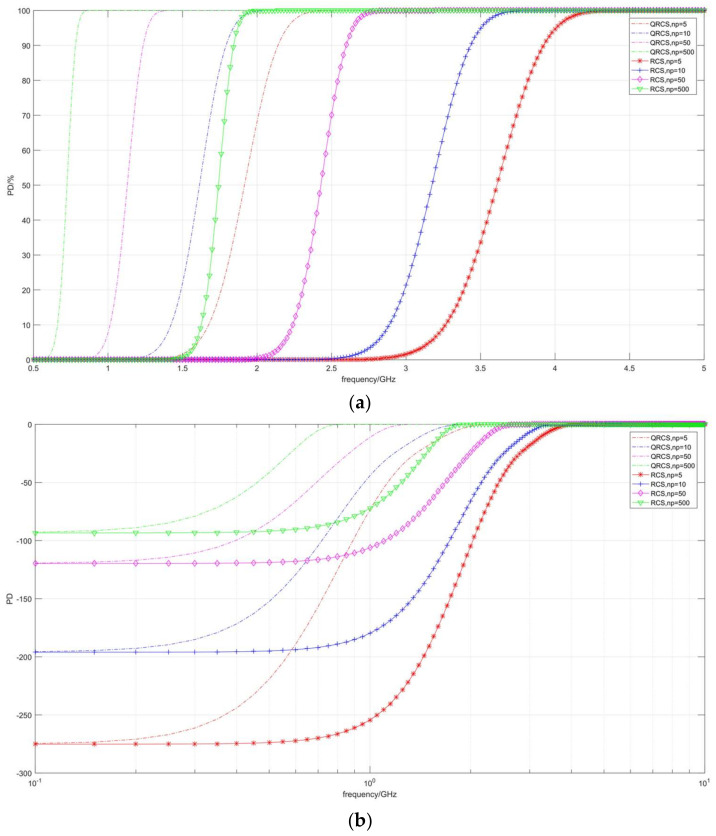
Correlation diagram of detection probability changing with frequency **(taking**
*n_p_***)**. (**a**) *P_D_*/%, (**b**) *P_D_*/dB.

**Figure 8 sensors-22-05944-f008:**
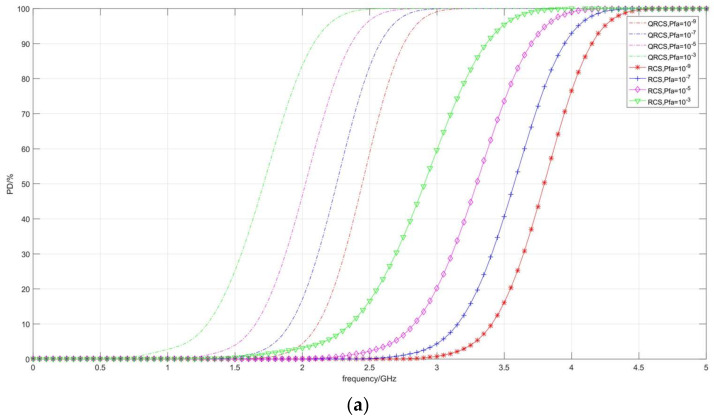
Correlation diagram of detection probability changing with frequency (taking *P_fa_*). (**a**) *P_D_*/%, (**b**) *P_D_*/dB.

**Figure 9 sensors-22-05944-f009:**
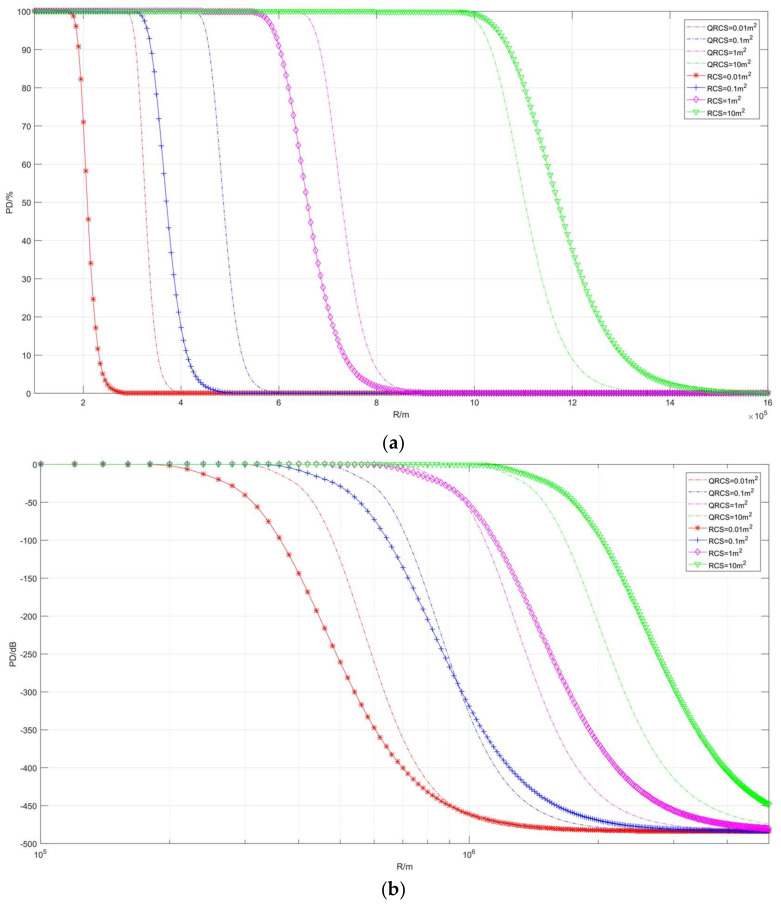
Correlation diagram of detection probability changing with distance (taking different RCS and QRCS). (**a**) *P_D_*/%, (**b**) *P_D_*/dB.

**Figure 10 sensors-22-05944-f010:**
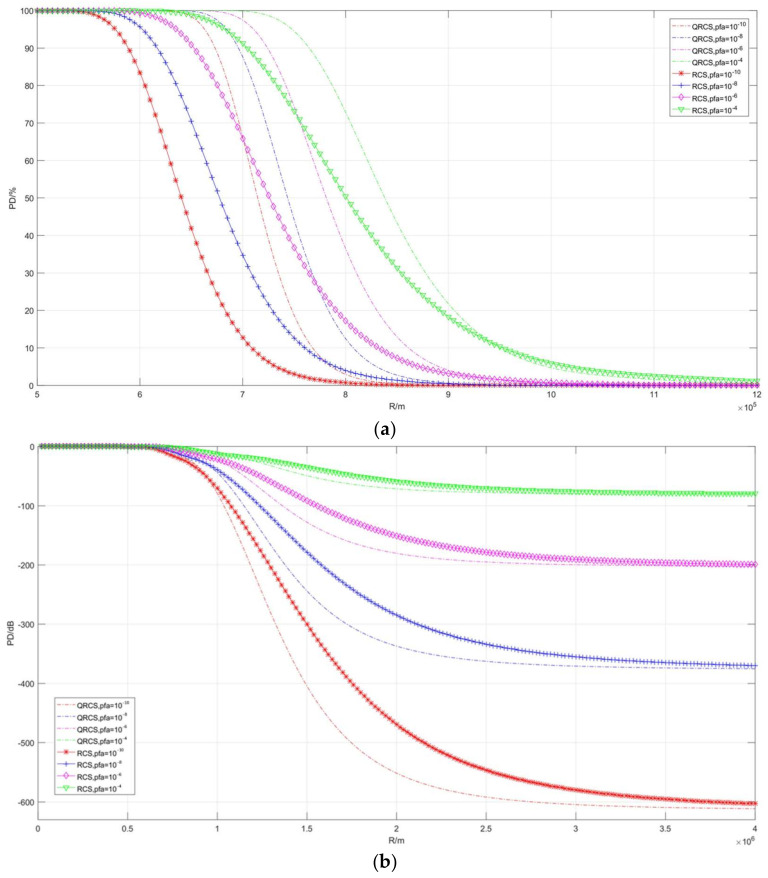
Correlation diagram of detection probability changing with distance (taking different *P_fa_*). (**a**) *P_D_*/%, (**b**) *P_D_*/dB.

**Figure 11 sensors-22-05944-f011:**
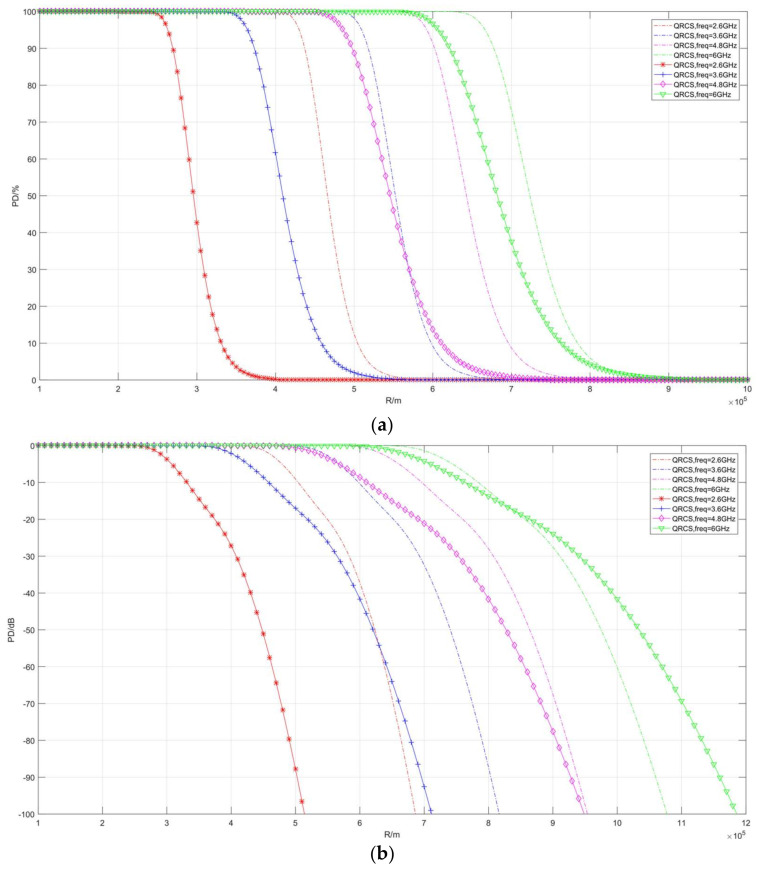
Correlation diagram of detection probability changing with distance **(taking different frequencies)**. (**a**) *P_D_*/%, (**b**) *P_D_*/dB.

**Figure 12 sensors-22-05944-f012:**
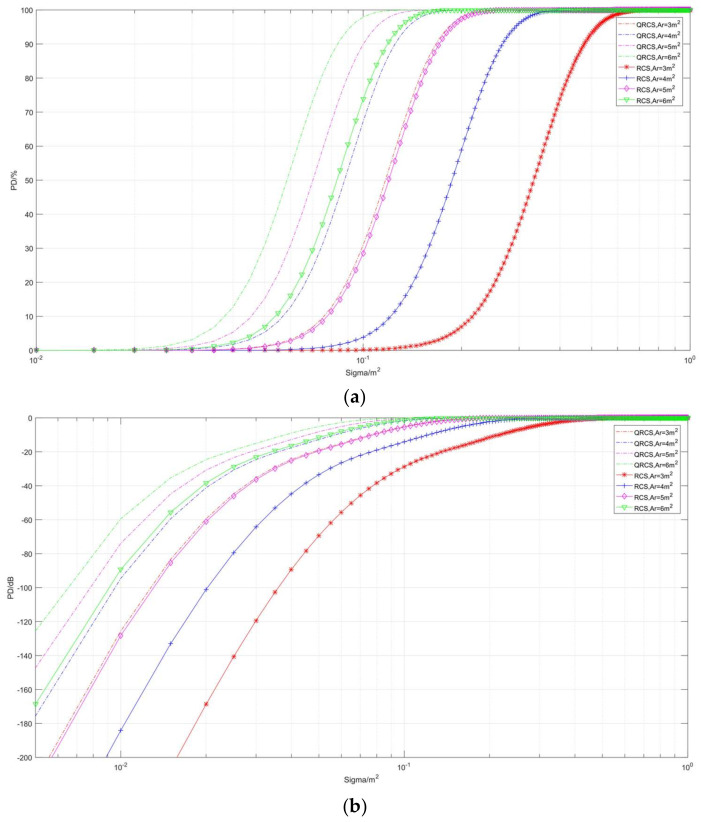
Correlation diagram of detection probability changing with RCS and QRCS (Taking different *A_r_*). (**a**) *P_D_*/%, (**b**) *P_D_*/dB.

**Figure 13 sensors-22-05944-f013:**
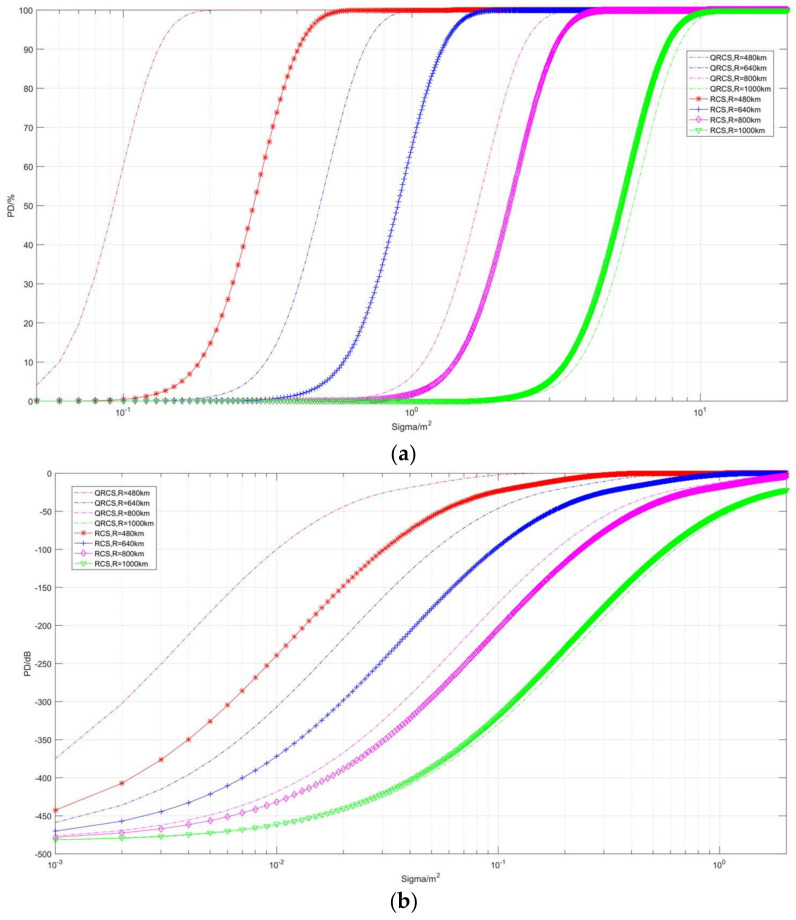
Correlation diagram of detection probability changing with RCS and QRCS (Taking different distances). (**a**) *P_D_*/%, (**b**) *P_D_*/dB.

**Figure 14 sensors-22-05944-f014:**
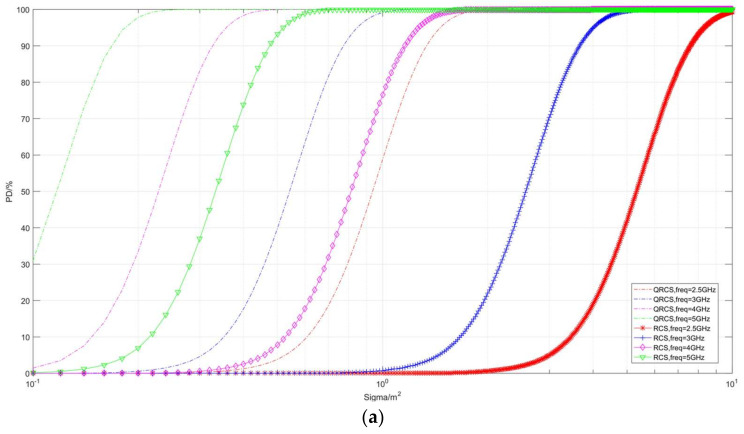
Correlation diagram of detection probability changing with RCS and QRCS (Taking different frequencies). (**a**) *P_D_*/%, (**b**) *P_D_*/dB.

**Figure 15 sensors-22-05944-f015:**
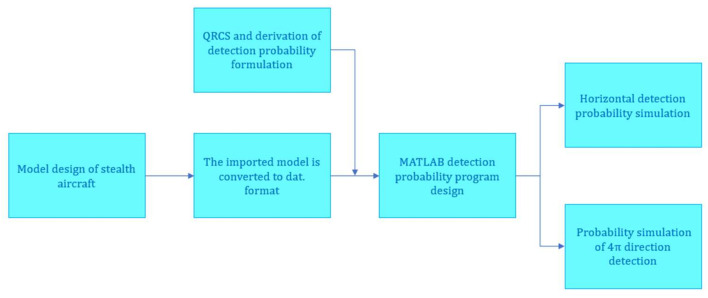
Simulation process.

**Figure 16 sensors-22-05944-f016:**
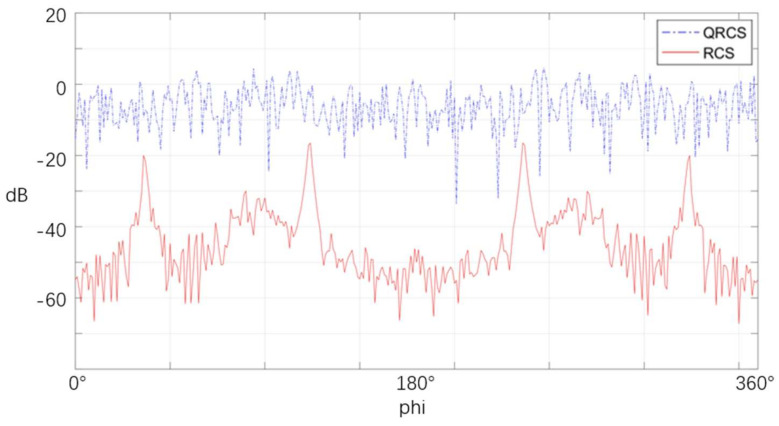
RCS and QRCS of the flying-wing stealth aircraft at various azimuth angles without pitch angle.

**Figure 17 sensors-22-05944-f017:**
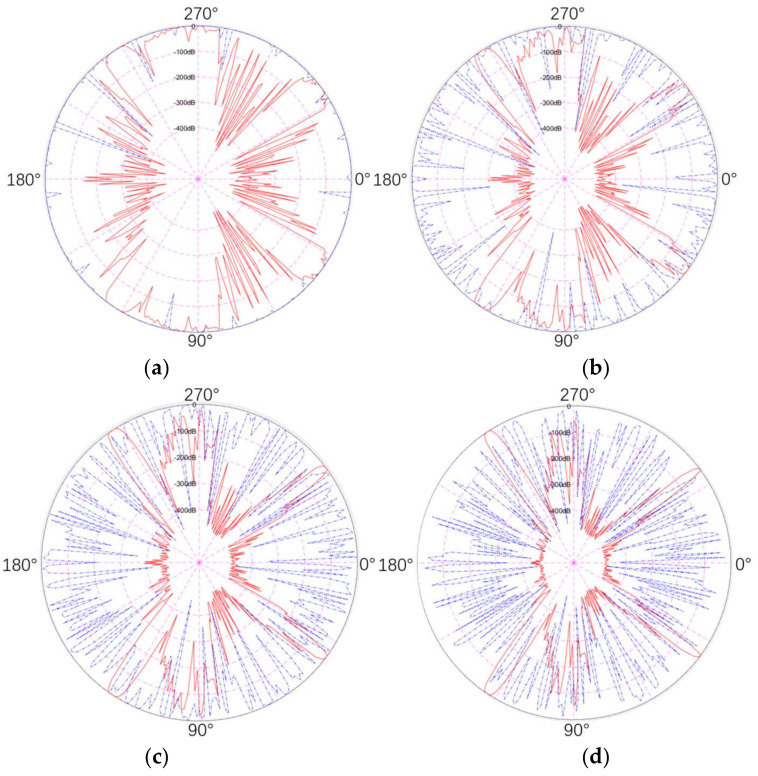
Comparison of detection probability between the quantum radar and the conventional radar at different distances (red line is *P_D_*, blue line is *P_DQ_*). (**a**) 600 km, (**b**) 800 km, (**c**) 1000 km, (**d**) 1200 km, (**e**) 1400 km, (**f**) 1600 km.

**Figure 18 sensors-22-05944-f018:**
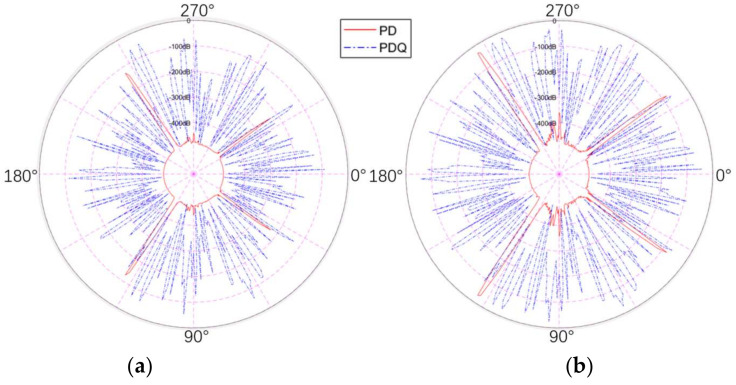
Comparison of detection probability between the quantum radar and the conventional radar at different aperture areas (PD in red lines and PDQ in blue lines). (**a**) 1.4 m^2^, (**b**) 2.6 m^2^, (**c**) 7.3 m^2^, (**d**) 10.5 m^2^, (**e**) 13.9 m^2^, (**f**) 21.5 m^2^.

**Figure 19 sensors-22-05944-f019:**
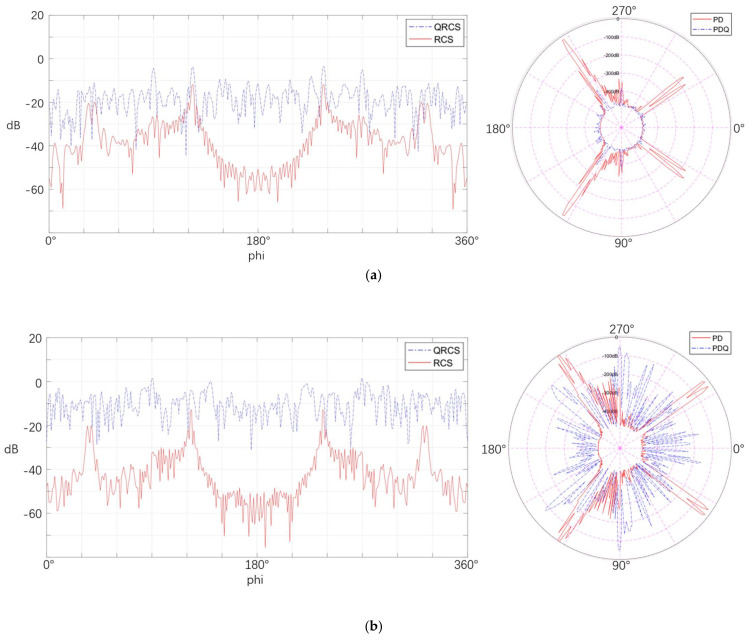
Comparison of detection probability between the quantum radar and the conventional radar at different frequencies. (**a**) 1.1 GHz, (**b**) 2.9 GHz, (**c**) 3.7 GHz, (**d**) 4.4 GHz, (**e**) 5.5 GHz, (**f**) 8.1 GHz, (**g**) 10 GHz.

**Figure 20 sensors-22-05944-f020:**
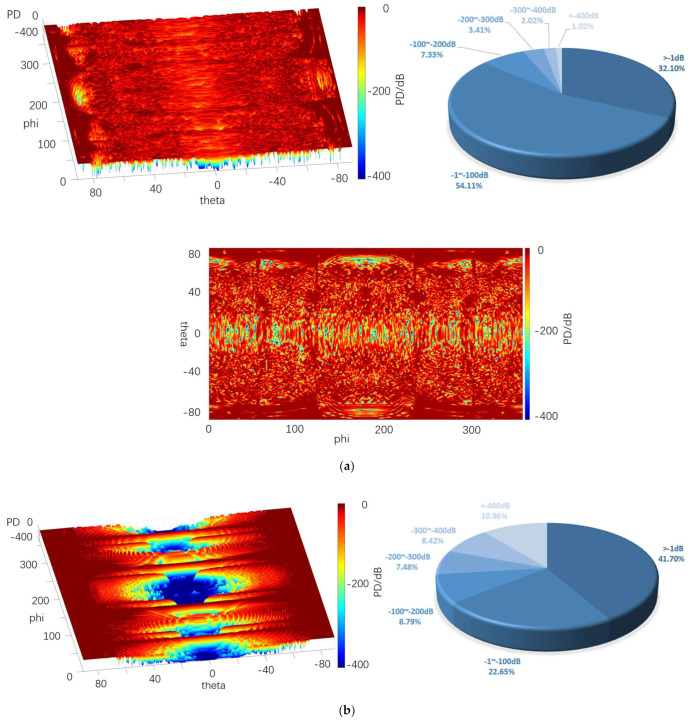
Probability distribution in the 4π direction for 1000 km. (**a**) Quantum detection probability, (**b**) Probability of conventional detection.

**Figure 21 sensors-22-05944-f021:**
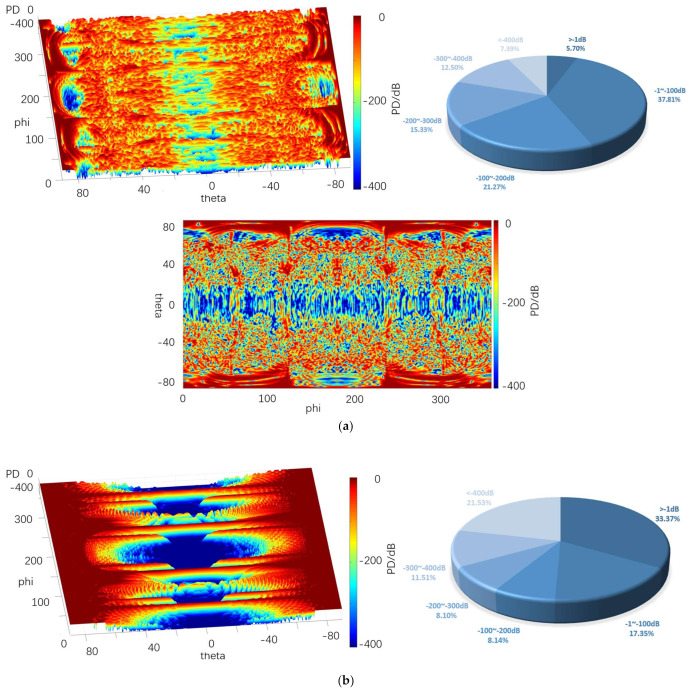
Probability distribution in the 4π direction for 1400 km. (**a**) Quantum detection probability, (**b**) Probability of conventional detection.

**Figure 22 sensors-22-05944-f022:**
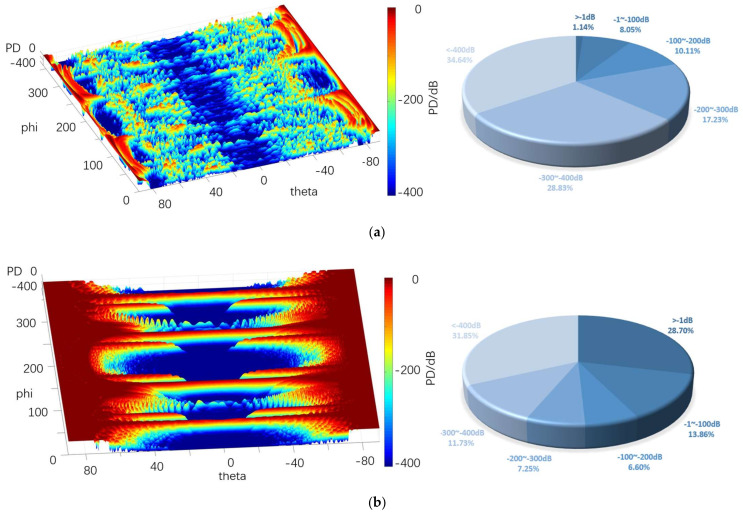
Probability distribution in the 4π direction for 1800 km. (**a**) Quantum detection probability, (**b**) Probability of conventional detection.

**Figure 23 sensors-22-05944-f023:**
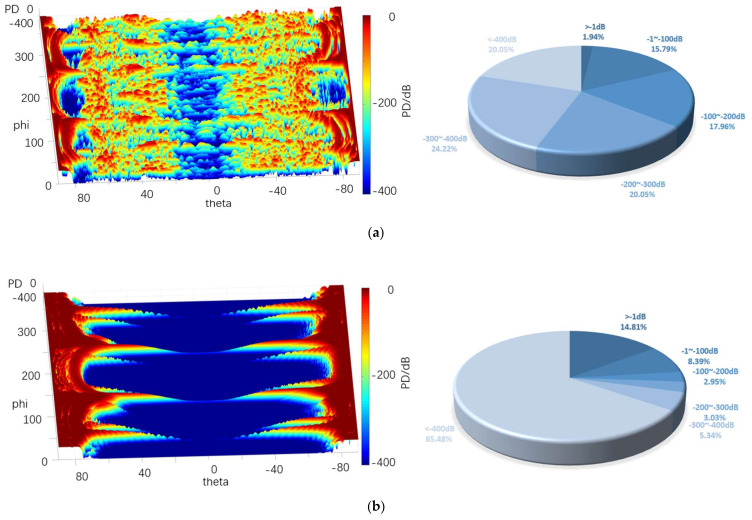
Probability distribution in the 4π direction of 0.6 m^2^. (**a**) Quantum detection probability, (**b**) Probability of conventional detection.

**Figure 24 sensors-22-05944-f024:**
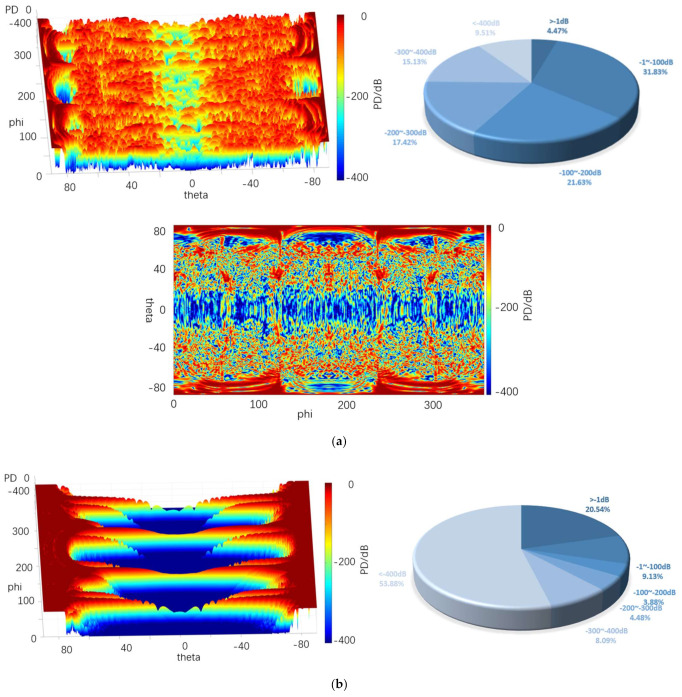
Probability distribution in the 4π direction of 2.7 m^2^. (**a**) Quantum detection probability, (**b**) Probability of conventional detection.

**Figure 25 sensors-22-05944-f025:**
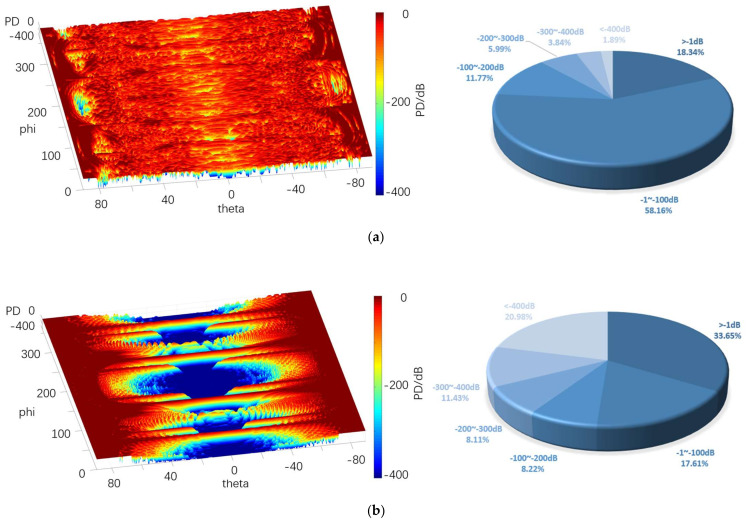
Probability distribution in the 4π direction of 7.3 m^2^. (**a**) Quantum detection probability, (**b**) Probability of conventional detection.

**Table 1 sensors-22-05944-t001:** Average quantum and conventional radar detection probability at different distances.

R	600 km	800 km	1000 km
*P_DQ_*/dB	−0.59	−4.00	−11.79
*P_D_*/dB	−8.99	−12.48	−14.43
R	1200 km	1400 km	1800 km
*P_DQ_*/dB	−28.23	−55.93	−243.02
*P_D_*/dB	−15.91	−18.33	−26.57

**Table 2 sensors-22-05944-t002:** Average detection probability of quantum and conventional radar at different aperture areas.

***A_r_*/m^2^**	1.4	2.6	7.3
**P_DQ_/dB**	−71.63	−42.63	−18.11
**P_D_/dB**	−147.34	−52.81	−19.58
***A_r_*/m^2^**	10.5	13.9	21.5
**P_DQ_/dB**	−14.70	−11.79	−8.16
**P_D_/dB**	−15.36	−14.43	−12.53

**Table 3 sensors-22-05944-t003:** The average detection probability of the quantum radar and the conventional radar at different frequencies.

	1.1GHz	1.8 GHz	2.9 GHz	3.7 GHz
*P_DQ_*/dB	−480.56	−372.28	−72.16	−48.86
*P_D_*/dB	−198.83	−40.03	−20.51	−17.22
	4.4 GHz	5.5 GHz	8.1 GHz	10 GHz
*P_DQ_*/dB	−16.16	−10.89	−3.92	−1.83
*P_D_*/dB	−14.94	−14.39	−11.49	−9.20

**Table 4 sensors-22-05944-t004:** The mean probability of 4π direction for different distances.

	1000 km	1400 km	1800 km
*P_DQ_*/dB	−3.59	−4.58	−5.24
*P_D_*/dB	−4.11	−11.4	−18.61

**Table 5 sensors-22-05944-t005:** Mean of different aperture areas in the 4π direction.

	0.6 m^2^	2.7 m^2^	7.3 m^2^
*P_DQ_*/dB	−15.84	−12.41	−4.55
*P_D_*/dB	−8.06	−6.69	−4.64

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
