# Peer review of "Study on Quantum Radar Detection Probability Based on Flying-Wing Stealth Aircraft"

_sensors, 2022, doi:10.3390/s22165944_

Round 1

Reviewer 1 Report

The manuscript proposes a new method to obtain the detection probability of the quantum radar. The parameters including RCS, QRCS, frequency, radar aperture area, and distances are varied. In general, the manuscript can be considered for publication in Sensors after addressing the following minor comments,

- The text in the Fig. 9-25 is too small to read.

- What is the full name of CATIA?

Author Response

Dear Reviewer:

Thank you very much for your recognition of the paper. I have made detailed modifications to the paper according to your opinions, and some pictures of the paper have been modified. Thank you again for your suggestions!

CATIA is short for Computer Aided Three-dimensional Interactive Application, it is a 3d modeling software.

Reviewer 2 Report

The authors have presented a very interesting article. However, there are many things lacking in the article. The abstract and the introduction has to be rewritten. The introduction does not give any substantial information on the articles contribution as compared to previous approaches. 

Secondly, there are no figures in the article, hence, the results are not appropriately corroborated. Hence, I would suggest that the article be properly presented before being considered for publication. 

Author Response

Dear Reviewer:

Thank you very much for your suggestions. I have made detailed modifications according to the problems you pointed out, and described the contribution of this paper compared with previous methods in the latter part of the abstract and introduction. In addition, when uploading files, there was a problem that pictures could not be displayed, which causing unnecessary trouble. Therefore, after the article is modified, I will send you a PDF version of the article, so that there will be no problem of missing data. Thank you for your understanding!

Reviewer 3 Report

Considering OAM can improve RCS?

Author Response

Dear Reviewer:

Thank you for your recognition of the article. The occlusion factor is considered to simulate the scattering characteristics of photon from the target in real situations, and to achieve accurate calculation of the quantum radar scattering cross section by excluding unirradiated areas.

Reviewer 4 Report

Unfortunately, my drawings did not open. I asked to send them separately, but I didn't receive anything. Therefore, I can only be responsible for the theoretical part. This part was done flawlessly, very good results were obtained, confirmed by analytical calculations. In this paper, for the first time, a new formula for the cross-section of a quantum radar is derived. A method for obtaining the probability of detection of a quantum radar is proposed. The influence of various variables on the probability of detection, characteristics of the probability distribution of radar detection is investigated. I can't say anything about the experimental part, since there are no drawings.

Author Response

Dear Reviewer:

Thank you very much for your recognition of the article, and I feel very sorry for not taking your feelings into consideration. After the article is modified, I will send you a PDF version of the article, so that such problems will not occur, thanks for your understanding!

Reviewer 5 Report

* It woud be gret to explain to the readers how the sidelobes is an advantage. Tis is an elaboration to the following sentence in the Introduction "They suggested that the sidelobe of planar QRCS was higher than that of RCS, so the quantum radar was confirmed to show the advantage of sidelobe".

Author Response

Dear Reviewer:

Thank you for your recognition and suggestions on the article. I have revised the article according to the problems you pointed out and corrected unnecessary explanations. Thank you again for your comments.

Round 2

Reviewer 2 Report

Great job from the authors.